# Multicomponent Training in Progressive Phases Improves Functional Capacity, Physical Capacity, Quality of Life, and Exercise Motivation in Community-Dwelling Older Adults: A Randomized Clinical Trial

**DOI:** 10.3390/ijerph20032755

**Published:** 2023-02-03

**Authors:** Emilio Jofré-Saldía, Álvaro Villalobos-Gorigoitía, Cristián Cofré-Bolados, Gerson Ferrari, Gemma María Gea-García

**Affiliations:** 1Instituto de Ciencias de la Salud, Universidad de O’Higgins, Rancagua 2841935, Chile; 2Facultad de Educación y Ciencias Sociales, Instituto del Deporte y Bienestar, Universidad Andres Bello, Santiago 7550000, Chile; 3Instituto del Deporte, Universidad de las Américas, Sede Providencia, Manuel Montt 948, Santiago 7500975, Chile; 4School of Physical Activity, Sport and Health Sciences, University of Santiago de Chile, Santiago 9170022, Chile; 5Faculty of Health Sciences, Universidad Autónoma de Chile, Providencia 7500912, Chile; 6Faculty of Sport, Catholic University of Murcia, 30107 Murcia, Spain; 7Health, Physical Activity, Fitness and Motor Control Performance Research Group (GISAFFCOM), Catholic University of Murcia, 30107 Murcia, Spain

**Keywords:** functional capacity, fitness, multicomponent training, older adults

## Abstract

Objective: To evaluate the effect of a multicomponent progressive training program (MPTP) on functionality, quality of life (QoL) and motivation to exercise (EM) in a group of older adults (OA) of a community. Methods: A total of 55 participants of 69.42 ± 6.01 years of age were randomized into two groups; experimental (EG:35) and control (CG:20), and subjected to 27 weeks of MPTP. Functionality (pre/post-intervention) was assessed using the Short Physical Performance Battery (SPPB), Time Up and Go (TUG), Walking While Talking Test (WWT), Manual Dynamometry (MD), Forced Expiratory Volume in the first second (FEV_1_), Sit and Reach (SR), Back Scratch (BS), and walk for 2 min (2 mST). QoL was assessed using the SF-36 questionnaire and EM using the BREQ-3. The Kolmogorov–Smirnov and Levene tests were applied. A two-way repeated measures ANOVA was applied. A significance level of *p* < 0.05 was accepted for all comparisons. Results: The EG compared to the CG improved in SPPB (ΔEG/CG: 29.67%/*p* < 0.001), TUG (ΔEG/CG: 35.70%/*p* < 0.05), WWT (ΔEG/CG: 42.93%/*p* < 0.001), MD (ΔEG/CG: 20.40%/*p* < 0.05), FEV_1_ (ΔEG/CG: 21.37%/*p* < 0.05), BS (ΔEG/CG: 80.34%/*p* < 0.05), 2 mST (ΔEG/CG: 33.02%/*p* < 0.05), SF-36 (ΔEG/CG: 13.85%/*p* < 0.001), and Intrinsic Regulation (ΔEG/CG: 27.97%/*p* < 0.001); Identified by regulation (ΔEG/CG: 9.29%/*p* < 0.05). Conclusion: An MPTP improves functionality, QoL and EM, and is a safe and effective method for community OAs.

## 1. Introduction

The demographic transition of recent decades has generated a significant increase in the life expectancy of the population, which has led to a higher proportion of older adults worldwide [1]. In the case of Chile, 16.2% of the population corresponds to people aged 60 and over, and the average life expectancy reaches 80.7 years. It is expected that by the year 2050, this group of people will reach 32% of the total population [2]. Aging is a physiological process that involves a gradual deterioration of the functional reserve of all body systems, especially the neuromuscular and cardiorespiratory systems, which largely determine the physical fitness of an individual [3]. In addition, physical inactivity has been shown to further accelerate the damaging effects of aging, increasing the risk of frailty, associated diseases, and premature mortality [4]. Therefore, inactive aging affects functional capacity and physical capacity, which has a negative impact on their ability to carry out basic activities of daily living, worsening the quality of life of the elderly (here understood as the perception that the individual has regarding their health, autonomy, independence, and satisfaction with life [5]). Faced with this situation, physical exercise (PE) turns out to be an essential coadjutant measure to attenuate the natural effects of aging and improve function levels in this population [6]. According to the recommendations made by the American College of Sports Medicine (ACSM), PE interventions should include the development of strength, cardiorespiratory fitness, balance, and flexibility [7]. Along this line, it has been shown that strength training in older people delays the effects of sarcopenia, and is able to attenuate the loss of the functional reserve of the organism [8]. In this sense, it has been shown that during the first weeks of strength training there may be increases of 10–30% and more in this physical quality in older people [9]. On the other hand, training programs focused on aerobic endurance produce central and peripheral adaptations that lead to increases in maximal oxygen consumption of 16 to 19% in older adults, which translates into a reduced risk of falls [10]. Finally, programs with static stretching exercises and full-range movements demonstrate significant improvements in lower back/hamstring flexibility (+25%), spinal extension (+40%), and leg and shoulder mobility in 70-year-old men and women [11]. Within the PE modalities, multicomponent training has been shown to be an effective strategy to improve various parameters related to function and general health in older adults [12]. Multicomponent training is a PE modality, in which various physical qualities (strength, cardiorespiratory endurance, flexibility, and balance) are developed with an equal volume distribution in the same session (approximately 60 min) [13]. However, older people who do not perform PE on a regular basis might not be able to perform physical conditioning sessions of very long duration at the beginning of a training program, due to their decreased functional and physical capacity [14]. This situation can generate a feeling of excessive tiredness, which can increase demotivation and cause a possible abandonment of the practice of PE [15]. For this reason, training programs of this type should be applied with a gradual progression that incorporates the work of physical qualities over the weeks [16]. In this context, the development of strength should be the basis of the training periodization to increase, in the first instance, the functional capacity and thus begin the development of activities that require greater physical effort [17]. This may have as a consequence increasing levels of motivation to perform PE, since older people would present lower levels of exhaustion, improving self-efficacy to carry out their daily activities. This would lead to increased levels of independence and autonomy [18]. In this way, adherence can be maintained and the dropout rates of PE programs by older adults can be reduced in order to improve the quality of life in dimensions such as physical function, vitality, emotional factors, and general health [19]. In this sense, there are no similar studies that apply a strength program in its early stages and that gradually integrate the other physical qualities to benefit physical function and quality of life. Therefore, the aim of this research was to apply a multicomponent training program distributed in progressive phases in order to assess its effects on parameters related to functional capacity, physical capacity, quality of life, motivation for exercise, and body composition in a group of older adults living in the community. The hypothesis contemplated for the current investigation was that multicomponent training distributed in progressive phases would be capable of improving functional capacity, physical capacity, quality of life, and motivation for exercise without predicting large changes in body composition.

## 2. Materials and Methods

### 2.1. Design and Participants

A total of 55 older adults with a mean age of 69.42 ± 6.01 (Me = 70; range = (60–80)) years old participated in this study. Further details on the specific characteristics of the sample can be found in Table 1 below. Specifically, the participants were selected using a probabilistic sampling technique through a simple random selection strategy [20]. This research was developed according to the CONSORT guidelines [21] and was written according to the Interventional Trials (SPIRIT) guidelines [22]. The subjects were randomly distributed into two groups by using specific software (https://www.randomizer.org, accessed on 31 July 2019), identified as a control group (CG) and an experimental group (EG), with 20 subjects assigned to the CG and 35 to the EG.

The selection of subjects was conditioned by the fulfillment of a series of prerequisites, to ensure that the subjects did present some type of contraindication to carry out physical activity, and that it would not limit their functional performance when following the training protocol in the case of being randomly selected to belong to that experimental group. For this, the exclusion criteria determined by the research group were: (i) not being able to move from one point to another without personal or technical assistance; (ii) presenting any type of contraindication for carrying out the multicomponent training to be developed as an intervention protocol in this research (such as muscle or joint injuries or fractures in the last three months; (iii) terminal illnesses; (iv) presenting pathologies identified with severe or terminal cardiovascular conditions; (v) pathologies associated with dementia, depression or Alzheimer’s [23].

All participants gave their informed consent in writing to participate in this study, which was approved by the ethics committee of the participating universities (code: CE101801). Finally, this study was developed in accordance with the ethical principles of Helsinki [24].

### 2.2. Sample Size

The calculations to establish the sample size were performed with the G*Power 3.1.9.4 software. The level of significance was set at α = 0.05. Consequently, the sample size (power analysis) revealed that 54 participants would have 95% power [25]. To avoid possible dropouts or the elimination of the data recorded due to the detection of an abnormal response or dropout, we decided to recruit a larger number of participants; for instance, the initial study sample consisted of a total of 102 subjects. These 102 subjects were randomly distributed into two groups, CG = 51 and EG = 51, as explained above using specific software (https://www.randomizer.org, accessed on 31 July 2019). After the planned 27 weeks of training, these groups gradually lost components, so that in the end, the sample was made up of a total of 55 participants, distributed into groups as explained in the participants section (Figure 1). In both groups, the loss of components is mainly due to logistical factors such as transportation problems, change of address, and, in some cases, the appearance of diseases that prevented them from attending the intervention center. However, in the CG the greatest cause of withdrawal was attributed to a loss of interest in participating, since no intervention was applied to this group.

### 2.3. Measures and Procedure

#### 2.3.1. Intervention Program

The description of the intervention follows the TIDieR checklist [26]. The EG carried out a 27-week multicomponent training program divided into 3 phases of 9 weeks each, where physical qualities were progressively worked together twice a week, on non-consecutive days during the morning session and in groups of 10–15 participants. In each phase, the development of a physical quality predominated, in order to progressively improve functional capacity and physical capacity, and lead to a higher quality of life related to health and motivation for exercise. The main objective of the first phase (Table 2) was to develop strength through variable resistance machines and overload exercises (i.e., elastic bands and medicine ball). This phase was subdivided into 3 blocks of 3 weeks each: Neuromuscular adaptation (block 1), muscular power (block 2), and muscular endurance (block 3). Each session lasted approximately 45 min at an intensity of 70–75% of 1RM and with light to maximal effort. The session consisted of 10 min of mobility and muscle activation as a warm-up, 25 min of strength exercises and 10 min of stretching as a cool-down. Next, the main objective of the second phase (Table 2) was to develop cardiorespiratory capacity through intermittent walking training in a room with dimensions of 20 m long by 10 m wide (walking routes were demarcated with colored cones). As in the first phase, the training was here also distributed in 3 blocks of 3 weeks each identified with static walking exercises (block 1), dynamic walking (block 2) and dynamic walking with changes of direction (block 3). Each session lasted approximately 50 min with a perceived exertion of 6–8 according to the Borg scale. The session consisted of 10 min of mobility and muscle activation as a warm-up, 10 min of muscular power exercises, 20 min of cardiorespiratory resistance, and 10 min of stretching as a cool-down. Finally, the main objective of the third phase (Table 2) was to develop balance and flexibility, by strengthening the stabilizing muscles and improving the range of motion (for example, through exercises with bosu, mini bosu, minitramp, and fitball). In turn, this phase was subdivided into 3 blocks of 3 weeks each: Balance/static flexibility (block 1), balance/dynamic flexibility (block 2) and balance/flexibility with double task (block 3). Each session lasted approximately 60 min with a perceived exertion of 6–8 according to the Borg scale. The session consisted of 10 min of mobility and muscle activation as a warm-up, 10 min of power exercises, 10 min of cardiorespiratory resistance, 20 min of balance exercises, and 10 min of stretching as a cool-down). Strength loads were individualized using the 10 RM test [27] per exercise every 6 sessions and titrated by character of effort (EC) [28]. Loads for cardiorespiratory work, balance, and flexibility were established using the range of perceived exertion (RPE) [29]. Both the EC and the RPE were explained at the beginning of the intervention and reviewed in each session to control the perceived exertion in each exercise. Participants’ attendance at the program was recorded throughout the intervention and at the beginning of each session. To favor the adherence of the experimental group to the program, the sessions did not continue work until muscular failure, thereby avoiding fatigue and muscular pain that could cause demotivation and abandonment [27]. In addition, the setting was adapted according to their tastes and preferences (for example, through the use of music, the attention of the research team to their concerns, and weekly coexistence meetings).

Older people assigned to the control group did not perform any training schedule. They only attended the measurements, and it was verified that they did not participate in other programs, through telephone contact with the director of the elderly department of the commune.

#### 2.3.2. Measures

This study was carried out between October 2019 and March 2020. Prior to data collection, participants attended a familiarization session for each test. During the familiarization session, the study participants completed the anamnesis regarding self-reported medical history, lifestyle habits and physical activity, the SF-36 questionnaire, and the BREQ-3 questionnaire. One week after familiarization, the dependent variables were tested as described below.

Tests to assess functional capacity, physical capacity, quality of life, and body composition were applied at weeks 1, 9, 18, and 27. The evaluations of these variables were applied in these weeks to assess the effects of each of the phases of the program. Motivation to exercise was assessed at weeks 1 and 27 to give a broader perspective of the effect of the entire program on this variable. These were performed between 8:00–12:00 h and there was a 5 min break between measurements. The evaluator in charge of the tests was blinded, so he did not know to which group each participant belonged. All participants were asked to maintain their normal daily routines and eating habits.

Next, the measures related to the dependent variables of this research will be detailed:

##### Body Composition

To evaluate this variable, the electrical impedance measurement technique was used, by using the OMRON HBF-514^®^ device, which provides an anthropometric profile based on the weight/height ratio, % fat/% muscle mass, and biological age. This device is a safe, reproducible and reliable tool to assess body composition in people up to 80 years old (Men r: 0.94-Women r: 0.89) [30]. As a complement, the abdominal perimeter was evaluated using SECA S201^®^ metric tape, which has an accuracy of 0.1 cm [31]. It was measured at the midpoint between the inner edge of the last rib intercepted with the anterior axillary line and the iliac crest, verifying that the person was not inhaling or having a forced expiration, and the result was recorded in centimeters [32]. Its use is valid and reliable for carrying out interventions in the community [31].

##### Functional Capacity

For the measurement of functional capacity, several tests were applied, such as: the Short Physical Performance Battery (SPPB), the 6 m Gait Speed (6 mGS) and the Timed Up and Go (TUG) tests.

The Short Physical Performance Battery (SPPB) is a test battery made up of different tests, which can cover different functional levels, and collects a score based on the results obtained in each test that can range between 0 and 12 points. For instance, if the result obtained is between 4 and 9 points, it suggests frailty [33]. This test is made up of three subtests, which are identified with activities that allow measuring the following: (i) balance, where the subtasks to be performed are those identified with balance tests with feet together, semi-tandem balance, and tandem balance; (ii) the push of the legs, identified with performing the action of getting up and sitting down from a chair five times as quickly as possible, and; (iii) walking speed at normal pace along 4 and 6 m. The SPPB is considered a valid and reliable method for measuring physical capacity in different populations (CCI: 0.87) [34].

Next, the measurements for the 6 mGS test were established through the time it takes for the person to travel a distance of 6 m, with a speed of less than 0.8 m/s being an indicator of frailty and risk of falls [35]. To do this, a researcher used a digital stopwatch, which made it possible to record the seconds it took for the participants to complete the course. This type of test is the simplest and most valid way of functional evaluation in the elderly (CCI: 0.87) [36].

In order to evaluate the mobility and function of the lower limbs, the Time Up and Go (TUG) test was used, in which a researcher, through the use of a digital stopwatch, measured the time in seconds that the person took getting up from a chair, walking 3 m, turning around, walking back at a normal pace and sitting in the same chair. This test has proven to be practical and reliable in this group of people (CCI: 0.95) [37]. In addition, it has been reported that a time ≥ 13.5 s is related to an increased risk of falling and frailty in older people [38]. In order to measure their ability to perform a double task (physical/cognitive), the Walking While Talking (WWT) test was applied, which measures gait speed under the condition of speaking while walking a distance of 6 m. For this, a researcher used a digital stopwatch to measure the time in seconds and the time spent was recorded. This instrument is reliable for measuring gait speed and cognitive ability in older people (CCI: 0.53–0.92) [39]. In this population, a lower gait speed and/or a greater number of stops and errors are considered markers of frailty and cognitive impairment [40].

##### Physical Capacity

In order to evaluate the components of physical capacity, various tests and instruments validated for use in older people were used. Therefore, to measure force, the CAMRY EH101^®^ model digital dynamometer was used. This allows recording the maximum isometric force of the upper extremity, which is a simple and reliable marker for this purpose (CCI: 0.95) [41]. The test was performed seated, with the elbow flexed at 90°. At the signal, the participant pressed the device for 5 s. Two attempts were made with both hands and the highest value was recorded. In older adults, values < 21 kg (men) and < 15 kg (women) are considered an indicator of frailty [42]. To evaluate physical resistance, the Two-Minutes Step Test was used, in which the greatest number of steps was recorded walking in place, each knee reaching an intermediate point between the patella and the anterior superior iliac spine, and the number of times the right knee reaches a given height was counted [43]. A score of less than 65 steps indicates that the subject has poor functional capacity and demonstrates high reliability for older adults (CCI: 0.91) [44]. To assess lung capacity, forced expiratory volume in 1 s (FEV_1_) was measured using the CMS-SP10^®^ manual spirometer, which has good reproducibility and a degree of correlation with lung function (CCI: 1.0) [45].

The participant inserted the mouthpiece into his mouth after maximal inspiration, and then exhaled and held for 6 s [46]. Values less than 1.5 L/s indicate poor lung function [47]. On the other hand, in order to measure the flexibility of the lower extremities, the Chair Sit and Reach test was used, which measures the flexibility of the biceps femoris and lumbar area. When sitting in a chair, the arms are extended to the tip of the foot, while the dominant leg is extended. The distance between the tips of the fingers of the hand and the tip of the foot was recorded, and the highest value of two attempts made was assigned [48]. This test is widely used in older adults, presenting high reliability (CCI: 0.99) [49]. To assess the flexibility of the upper limbs, the Back Scratch test was used, which measures the flexibility of the shoulder joint when trying to bring both hands towards the middle of the back. The result corresponds to the distance between the tips of the middle fingers of both hands, registering the distance as negative if the fingers do not touch, and positive if the fingers overlap. In the event that the fingers only touch, a zero score is assigned [50]. The test demonstrates high reliability in older adults (CCI: 0.99) [48].

##### Quality of Life

Quality of life was assessed using the SF-36 health questionnaire, which represents a generic scale that provides a profile of health status that can be applied to the general population [51]. The questionnaire is composed of a total of 36 items grouped into 8 different scales or dimensions: (a) physical functioning (composed of 10 items), (b) physical performance (composed of 4 items), (c) bodily pain (composed of 2 items), (d) emotional performance (composed of 3 items), (e) mental health (composed of 5 items), (f) vitality (composed of 4 items), (g) general health (composed of 5 items), and (g) social functioning (composed of for 2 items). The SF-36 scales are ordered so that the higher the score, the better the state of health [52]. The CCI are between 0.7–0.9 for this instrument and it is applicable to the general population, being valid for research and clinical practice [53]. The SF-36 is one of the most evaluated and often used quality of life questionnaires [54].

##### Exercise Motivation

Motivation for physical exercise was evaluated through the Exercising Behavior Regulation Questionnaire (BREQ-3), which is headed by the statement “I do physical exercise...” in 23 items: Four for intrinsic regulation, four for integrated regulation (“because it is in accordance with my way of life”, “because I consider physical exercise to be part of me”, “because I see physical exercise as a fundamental part of who I am”, “because I consider that physical exercise is in accordance with my values”), three for identified regulation, four for introjected regulation, four for external regulation, and four for amotivation [55]. The response format used was assessed on a Likert scale from 0 to 4, where 0 corresponds to total disagreement and 4 to total agreement [56]. It represents a valid questionnaire to be applied in older adults, since it is a reliable measurement instrument to measure the regulation of behavior underlying the self-determination theory in the exercise domain [57], demonstrating a CCI = 0.70–0088 [55].

#### 2.3.3. Statistical Analysis

The descriptive data of the sample are presented through the values of the mean and the standard deviation of the mean. The parameters of normality and homogeneity of the sample were verified through the Kolmogorov–Smirnov and Levene tests, respectively. Next, a two-way repeated measures analysis of variance was applied with TIME (before and after the application of the intervention program) and GROUP (CG vs. EG) as factors in order to analyze the possible effects and changes related to the training on each dependent variable for those variables with normal behavior. Within this statistical test, the Bonferroni post hoc test was applied when necessary to explore the possible differences between each of the two conditions. Finally, the equality of variance–covariance matrices was checked using the Box statistic. In the case of variables with non-normal behavior, the Friedman statistical test of measures was repeatedly applied with post hoc through Wilcoxon. The effect size (ES) was estimated by calculating the partial eta-squared (ŋP^2^) [<0.01 (small); >0.06 (medium) and >0.14 (large) effect] [58] and Cohen’s d [<0.2 (small); >0.5 (medium) and >0.8 (large) effect]. Finally, an independent samples t-test or Mann–Whitney U test, as appropriate, was applied to assess the prior differences in the descriptive scores of the sample according to the grouping of the subjects in CG or EG. A significance level of *p* < 0.05 was accepted for all statistical comparisons. Data analysis was performed by using SPSS software (IBM Corp., Armonk, NY, USA) for Windows, version 25.0.

## 3. Results

Table 1 shows the descriptive characteristics of the participants in each of the groups. As shown by the mean values for these characteristics, there were no significant differences in the values for the subjects depending on the assigned group (CG vs. EG).

Table 3 below shows the baseline and post-training values obtained for the variables related to the measurements of abdominal perimeter, functional capacity, and physical capacity in the CG and EG. Two-way repeated measures analysis revealed no significant effect of time on waist circumference values as a function of assigned training group (F_1_ = 1.140, *p* = 0.291, ŋP^2^ = 0.021). However, when studying the results obtained through the Bonferroni post hoc test, it was possible to verify that for the EG there were significant differences in these scores (F_1_ = 7.742, *p* = 0.034, ŋP^2^ = 0.082). More specifically, after submitting older adults to EM, it was possible to observe how the mean score decreased from 93.83 ± 12.08 cm to 92.05 ± 11.52 cm.

In this same table, the two-way repeated measures analysis allowed us to verify that for certain tests related to the functional capacity of older adults, a significant main effect of time was revealed. More specifically, the tests were: SPPB (χ2 = 89.61, *p* = 0.000, d = 0.352), tandem balance (F_1_ = 18.396, *p* = 0.000, ŋP^2^ = 0.261), speed 4 m (F_1_ = 30.783, *p* =0.000, ŋP^2^ = 0.367), standing 5 times on a chair (GUS5) (F_1_ = 13.135, *p* = 0.001, ŋP^2^ = 0.199), TUG (F_1_ = 8.197, *p* = 0.006, ŋP^2^ = 0.134) and gait speed in 6 m (F_1_ = 39.939, *p* = 0.000, ŋP^2^ = 0.390). In relation to the SPPB, when comparing the scores obtained by the EG after the application of the MPTP, it was possible to verify how the final general score obtained by this EG was higher (11.80 points) than that obtained by the CG (9.10 points) (*p* = 0.000). In fact, if the results obtained individually for each of the tests that make up the SPPB are considered, it can be seen that there were significant differences in the mean scores obtained for the tandem balance tests (T) (*p* = 0.000), the 4 m walk speed test (4 mGS) (*p* = 0.000) and the test to get up and sit on a chair five times (GUS5) (*p* = 0.000). In each and every one of the tests, the results for the GE showed an improvement in the scores obtained with a percentage of 64.74% for the T test, 33.40% for the 4 mGS test, and 36.01% for the GUS5 test. In this same interaction, it was possible to observe how, when studying the scores obtained separately by each intervention group before and after the MPTP, both the EG and the CG presented significant differences in the general mean scores obtained for the SPPB (*p* = 0.000). After these older adults were subjected to the MI protocol, the score obtained in the SPPB was 11.32% higher than the original or baseline score for the EG, while for the CG the result obtained presented a decrease of 3.70%. In fact, if the results obtained individually for each of the tests that make up the SPPB are taken into account, it can be seen that there were significant differences in the mean scores obtained for the tandem balance tests (T) (*p* = 0.000), the 4 m gait speed test (4 mGS) (*p* = 0.000) and the test to get up and sit on a chair five times (GUS5) (*p* = 0.000) in the case of the EG. In each and every one of the tests, the results for the EG showed an improvement in the scores obtained with a percentage of 9.03% for the T test, 15.07% for the 4 mGS test and 24.39% for the GUS5 test. Meanwhile, for the CG, when comparing their mean scores in the tests that make up the SPPB, significant differences were found in the scores for the semi-tandem balance test (ST) (*p* = 0.019) and 4 mGS (*p* = 0.000); for the ST test there was an improvement of 11.80% and in the 4 mGS test the score worsened by 21.6%.

Finally, this significant interaction of time on the functional capacity tests also presented significant differences in the complementary TUG (*p* = 0.000) and 6 mGS (*p* = 0.000) tests. As in the previous cases, the EG in these tests improved its results compared to the scores obtained by the CG after the end of the intervention period through this MPTP, with percentages of improvement being 35.70% for the TUG test (EGM_Post = 7.67 sg versus CGM_Post = 11.92 sg), and 98.15% for the 6 mGS test (EGM_Post = 1.87 mts/sg versus CGM_Post = 0.94 mts/sg). As in the previous case, when comparing the mean baseline scores with those after the MPTP between groups, it was observed that in both cases, both the EG and the CG presented significant differences (*p* < 0.05). More specifically, the EG experienced an improvement of 0.066 mts/sg, which represents a 54.61% increase in movement speed for the 6 mGS test and 7.37% for the TUG test; while in the case of the CG, these percentages meant a worsening of their results in the two physical capacity evaluation tests, which indicates a detriment of their movement speed of 16%07 in the 6 mGS and an increase of 13.3% in the time used for the development of the TUG test. On the other hand, these same two-way repeated measures analyses allowed us to verify that for certain tests related to the physical capacity of older adults, a significant main effect of time was revealed. More specifically, the tests were: hand grip strength (HG_M) (F1 = 6.635, *p* = 0.013, ŋP^2^ = 0.111), 2 mST (F1 = 27.433, *p* = 0.000, ŋP^2^ = 0.341), Back Scratch (F1 =13.46, *p* = 0.001, ŋP^2^ = 0.230), FEV_1_ (F1 = 3.728, *p* = 0.059, ŋP^2^ = 0.059), and dual task (χ2 = 77.183 = 90.145, *p* = 0.000, d = 0.36). More specifically, when comparing the mean results for these physical capacity tests, it was possible to verify that for the five tests, the mean scores were better for the EG than in the CG. The EG presented levels of improvement of 20.40% in HG_M, 76.14% in the 2 min step test, 21.37% in the FEV_1_ test, 80.34% in the Back Scratch test, and 42.93% in the dual task test, compared to those of the GC. In addition, when comparing the results between groups, the EG presented significant differences in the mean scores obtained in the 2 mST (*p* = 0.000), and the Sit and Reach flexibility tests (*p* = 0.000) and Back Scratch (*p* = 0.000). Specifically, in these three physical capacity tests, the results showed an improvement of 33.02% for the 2 min step test, and 51.70% and 82.77% in the case of the Sit and Reach and Back Scratch tests, respectively. Finally, for the EG, the results showed significant differences only in the dual task test (*p* = 0.000), where a decrease in the time used to complete this evaluation was recorded.

Figure 2 below shows the most important results according to the type of test performed by capacity and physical capacity before and after MPTP.

Table 4 shows the baseline and post-training values obtained for the variables related to the measurements of quality of life and motivation for exercise in the CG and EG. More specifically, the two-way repeated measures analysis revealed a significant effect of time on the mean values obtained for the SF-36 test in each and every one of its dimensions (Physical Function (PF): F1 = 10.931, *p* = 0.002, ŋP^2^ = 0.174; Physical Role (PR): F1 = 15.598, *p* = 0.000, ŋP^2^ = 0.231; Body Pain (BP): F1 = 8.574, *p* = 0.005, ŋP^2^ = 0.142; General Health (GH): F1 = 7.498, *p* = 0.008, ŋP^2^ = 0.126; Vitality (V): F1 = 10.903, *p* = 0.002, ŋP^2^ = 0.173; Emotional role (ER): χ2 = 53.08, *p* = 0.000, d = 0.245; Social function (SF): F1 = 8.410, *p* = 0.005, ŋP^2^ = 0.139, and Mental Health (MH): F1 = 18.572, *p* = 0.000, ŋP^2^ = 0.263); while for the BREQ-3 test related to the parameter of motivation for physical exercise, a significant effect was only observed in the dimension of intrinsic regulation (Intr.R.) (χ2 = 77.183, *p* = 0.000, d = 0.155). On the other hand, even though the general results did not show a significant interaction of time over identified regulation (Id.R.) (χ2 = 88.032, *p* = 0.000, d = 0.146) and demotivation (Des) (F1 = 0.706, *p* = 0.405, ŋP^2^ = 0.013), when studying and comparing the means obtained by each of the intervention groups, significant differences were observed, as can be seen in Table 4.

Finally, Figure 3 shows these significant differences between groups after applying the MPTP program in more detail.

## 4. Discussion

This study aimed to assess the effects of an MPTP program divided into progressive phases on functional capacity, physical capacity, quality of life, exercise motivation, and body composition in older adults living in the community. To the best of our knowledge, this is the first intervention study that analyzes the effectiveness of an MPTP that is assembled in progressive blocks to gradually develop physical capacities thanks to the increase in intensity and volume of work over the weeks. This training program should be safe for the elderly who live in the community, since, as the scientific literature argues, training programs at these ages must comply with a series of specific characteristics, such as not continuing exercise until muscular failure, plenty of low/medium intensity work, and exercises and work methodologies that do not generate excessive fatigue or muscle pain associated with the demands of training that could discourage and abandon the practice of physical activity [27,29]. This type of design seeks that older people not only reduce their reticence towards the practice of PE, but also increase the participation and adherence of this type of group in PE programs [59,60], since this has repercussions on improvements for their health and quality of life [61]. In this sense, the PE program proposed here could meet these requirements, as will be seen later from the results obtained. In addition, this type of intervention registered a high average attendance rate, with 90% participation, and during its duration, no type of injury was reported, which is a guarantee of the possible effectiveness and adequacy of this type of program of PE for the target population of this research.

The anthropometric variables did not show any changes at the end of the intervention, except for the abdominal perimeter in the EG. This could be explained since the training program was not aimed at generating changes in body composition, since it had a low weekly frequency (2 days a week) and also, in its last phases, strength and cardiorespiratory training were considerably reduced in relation to the total volume of the session [62]. The changes in the abdominal perimeter could be explained because the scheduled RT is capable of reducing the abdominal circumference in older adults and in this case, the older adults participating in the program increased their usual energy expenditure [63]. It should be noted that this program was developed mainly with sets that were not close to failure, which could be a reason why there were no changes in muscle mass or fat percentage. However, it is important to note that the decrease in abdominal girth was a consequence of the training program and not an end in itself. Along these lines, it has been described that a smaller waist circumference is related to a better state of health in the general population [64].

Regarding functional capacity, the main findings of this research showed that an MPTP distributed in progressive phases can improve functional capacity evaluated by the SPPB, TUG and 6 mGS tests. More specifically, when referring to the results recorded for the SPPB, it was possible to verify how the EG improved their scores after applying the PE program. These improvements in the functional capacity of the EG could be due to the very design of the MPTP program, since it has work and exercise methods that follow the recommendations of clinical guidelines recognized worldwide for the prescription of physical exercise, such as the American Heart Association (AHA) [65], the American College of Sports Medicine [66], the National Strength and Conditioning Association (NSCA) [17], and the World Health Organization [67], all of which promote physical exercise programs that are mainly oriented to the development of the functional capacity of the elderly around health. More specifically, referring to the individual tests that make up the SPPB, it was found that the participants improved their records after applying the MPTP in the individual 4 mGS and GUS5 tests. In addition, the EG also showed improvements in the records obtained in the 6 mGS and TUG tests. These results are similar to those found in other studies where an MPTP was developed [68,69,70]. Although the studies mentioned above present a program design similar to the present research, this study has the advantage of developing work with a medium to low effort character (EC) and a lower total number of repetitions per series, which could be translated into an improvement in the quality of training [71], a complete recovery between sets and training sessions, and a decreased risk of injury [72], as evidenced throughout the training program. In addition, the gradual incorporation of strength, cardiovascular resistance, balance, and flexibility work by phases and progressive blocks, allowed the gradual development of functional capacity to perform sessions of longer duration and with greater variety of activities with better performance compared to conventional multicomponent training, which focuses on the development of all physical qualities in a single session [73]. Therefore, a multicomponent training modality with progressive characteristics has been shown to be effective in improving functional capacity, which could mean better performance at basic and instrumental activities of daily living [74].

Contrastingly, when comparing the scores obtained for the EG and the CG after the application of the training period, it was possible to verify that there were significant differences in the results obtained for the SPPB at a general level, and more specifically for the tests of balance in tandem, 4 mGS, and GUS5. Along the same line, the results obtained supported differences for the TUG and 6 mGS tests. These scores show that the application of an MPTP program allows not only a delay in the negative effects of aging associated with the lack of physical exercise at these ages, but also an improvement in the levels of functional capacity for the EG [75]. This fact agrees with the results found in other investigations that apply the PE program to this population group [75,76]. In fact, besides reflecting the differences between EG and CG, our results also show how the lack of PE causes a detriment to the functional capacity of the CG after those weeks, as reflected in the scores obtained in the 4 mGS or TUG tests. As the specific literature on the subject highlights, these tests are of special importance since they are associated with greater dynamic balance, greater walking speed, and less risk of falls, which is a clear marker of greater functional capacity and autonomy [77]. I is true that for the CG no differences were found or a slight improvement was observed in other key tests, such as balance in ST or the 6 mGS. This could be due to the fact that the study participants were already familiar with the protocol of these tests, and so the intention to perform them was with greater effort compared to the pre-test [78]. Regarding physical capacity, improvements were observed in muscle strength (Manual Dynamometry), walking capacity (2 mST), lower limb flexibility (SRT) and expiratory strength (FEV_1_), compared to the control group at the end of the intervention. These improvements could be explained because the physical qualities were developed in a periodized and progressive way. However, strength training was the basis of the program, which allowed the functional and physical capacity of the participants to be gradually built, making it possible for them to carry out activities of greater duration and complexity at the end of the intervention. Regarding the improvements in muscle strength, these could be due to the fact that phase 1 of the program had a progressive design that respected the principles of strength training [79,80]. That is, to start with a block of neuromuscular adaptation that improves intra- and intermuscular coordination, and thus prepare the body to perform work with greater speed of concentric execution (block 2,muscular power), and finally achieve greater intensity efforts (block 3, muscular endurance). It is important to mention that these improvements were maintained until the end of the program, despite the fact that the volume of work for strength training decreased in the following phases. This could be explained because phase 1 followed a linear training model, which has been shown to be an effective method for maintaining strength levels and functional performance in older adults [81]. Therefore, a higher hand grip strength represents an indicator of better muscle function and physical performance in activities of daily living [8]. In relation to the results obtained in the walking capacity test (2 mST), the improvements could be explained due to the progressive characteristics of the intervention, since developing strength in the first phase allowed the consolidation of functional capacity. Because of this, it was possible to develop works of greater duration in phase 2, due to better performance of the lower limbs, which is associated with an increase in walking capacity and speed [82]. On the other hand, the characteristics of phase 2 sought to develop cardiorespiratory capacity through static and dynamic intermittent walking, which is a method used to develop this physical quality in older adults [83]. Hence, a higher walking capacity represents an indicator of functional performance against daily tasks that require more effort, such as climbing stairs, shopping or crossing the street [84]. Regarding FEV_1_, the improvements found at the end of the intervention could be explained by the fact that, starting from phase 2, longer activities began to be progressively worked on, which allowed a greater development of cardiorespiratory fitness [85]. In addition, in phase 3, the work of all the physical qualities in the same session was incorporated, which configured the multicomponent training in its entirety, despite the fact that there is a study by Roldan et al., showing that this training modality was not a sufficient stimulus to improve this indicator [86]. In relation to our results, a higher FEV_1_ represents a better diagnosis of lung function in older adults [87]. It should be noted that there is limited evidence related to the effects of multicomponent training on FEV_1_ in older adults living in the community. For this reason, our study could help clarify the knowledge about the effects of these exercise modalities on lung function in older adults. Regarding the improvements in flexibility in the lower limbs (SRT), they could be explained due to the neuromuscular and physical performance adaptations achieved in phase 1 and 2. These improvements could be related to the decrease in antagonistic coactivation and the increase in the reciprocal inhibition reflex, which would improve control and amplitude of movement [88]. On the other hand, the characteristics of phase 3 sought to develop flexibility through a range of motion work and static/dynamic stretching, which are methods used to develop this physical quality in older adults [7]. Greater flexibility in the lower limbs is associated with a greater amplitude and frequency of stride, which allows daily activities related to locomotion to be performed more efficiently [11].

Health-related quality of life was assessed using the SF-36 questionnaire. At the end of the intervention, the experimental group presented improvements (higher scores) in the dimensions of physical function, physical role, general health, and vitality. These findings can be explained because phase 1 aimed at developing strength around functional capacity. In relation to this objective, it has been shown that strength training increases functional capacity and ultimately translates into an improvement in the perception of health-related quality of life [89]. Specifically, in this phase changes were observed in the SPPB score and Time Up and Go, which are considered significant predictors of greater autonomy in basic and instrumental activities of daily living, which translates into higher levels of independence and health-related quality of life in older adults [90]. In addition, once the strength phase was over, the work on cardiorespiratory resistance, balance, and flexibility was progressively incorporated. These are qualities that are linked to better physical capacity and dexterity, and are associated with an increase in vitality when performing activities of greater intensity and duration [91]. Another dimension that presented improvements at the end of the intervention was the emotional role, which could be explained by the fact that performing two strength training sessions per week may be the most beneficial for the emotional state of older adults [92]. When comparing both groups at the end of the intervention, differences were found in the dimensions of physical function, physical role, body pain, general health, vitality, emotional role, social function, and mental health. These findings could be explained by the fact that training programs with a frequency of two sessions per week, with emphasis on the development of different physical qualities in a progressive manner, are effective in improving health-related dimensions of quality of life regarding the social function of older adults. Especially if they are carried out in groups of 10 to 15 people, as has been the case in this intervention [93]. Furthermore, these findings are similar to those reported in other studies where an MPTP has been applied, in which health-related quality of life was also assessed [94,95]. Our research differed from previous studies by generating changes in quality of life with a lower weekly frequency (two sessions per week), lower volume (lower rep range) and lower intensity (moderate to low EC). In addition, prior to longer sessions, exercises aimed at improving gait through intermittent sessions were performed, which favored the development of more complex and prolonged tasks with a lesser sense of fatigue and greater functionality. In this sense, it has been described that older adults with better physical function can perform their activities of daily living with more vigor, which is associated with a higher health-related quality of life [96].

Motivation for exercise showed improvements in the dimension’s intrinsic regulation, identified regulation (increases in both variables) and demotivation (decrease in the score of this variable), in the experimental group at the end of the intervention. These findings could be explained by the fact that physical exercise has been widely shown to enhance positive health outcomes, improve willingness to engage in physical activity in general, and improve quality of life in this population [97]. On the other hand, when comparing the EG with the CG at the end of the intervention, differences were only found in the dimensions of intrinsic regulation and identified regulation. The observed results can be explained by the fact that physical training programs have been shown to have psychological and behavioral benefits in older adults, such as a better general mood [98]. Within this context, exercising favors enjoyment and satisfaction, and also allows us to assess the importance of having regular physical activity as a habit [99]. Within the scientific literature, there are no interventions where an MPTP is applied in which, at the end of the intervention, the dimensions related to motivation for exercise that the BREQ-3 questionnaire measures are evaluated. Our intervention improved the internal needs of competence and self-efficacy in the development of a task, as is the case of the voluntary practice of physical exercise and enjoyment of life, which are related to an increase in independence and autonomy [100]. Motivation is a very relevant psychological factor in this population and may be the driving force for older people to participate in activities related to active and healthy aging [19].

The strengths of this study are mainly related to the design of the multicomponent program, since by applying it in progressive phases, it is possible to gradually rebuild the autonomy and independence of the elderly, which could be diminished as a result of the high levels of sedentary lifestyle present in this population. Another strength is that exercise was not continued to muscle failure, and so the feeling of fatigue and exhaustion within the session and throughout the program was low. This resulted in a higher rate of adherence and motivation. The limitations found in this study are related to the size of the sample, so it is necessary to replicate the study with a larger number of participants. Another possible limitation of this study is that it has a quasi-experimental design. Therefore, characteristics inherent to this type of approach must be assumed, such as the lack of control over other variables that can influence physical functional improvements and quality of life. These types of variables could be feeding control, stress levels and/or socioeconomic problems, which can influence and cause variations in the results obtained.

## 5. Conclusions

In conclusion, our findings indicate that an MPTP improves functional capacity, physical capacity, quality of life, and motivation for exercise, compared to older adults who do not perform PE. However, more research is required to clarify which training methodology can have a greater impact on the variables considered in this research, in order to find practical strategies that allow improving the quality of life related to health in this population. A progressive MPTP methodology seems to be an effective and a safe strategy to improve the functional capacity of older adults, but more studies are required to verify its impact on the functionality of basic and instrumental activities of daily living. As a future line of research, different progressive training methodologies should be applied in this population, in order to be able to assess which training modality is the most effective in favoring the improvement of physical functional capacity, quality of life, and motivation for exercise in older people. In addition, the development of this type of intervention could be promoted in a younger population, in order to implement the PE as a prevention strategy for the conditions of aging with sedentary behavior at earlier ages.

## Figures and Tables

**Figure 1 ijerph-20-02755-f001:**
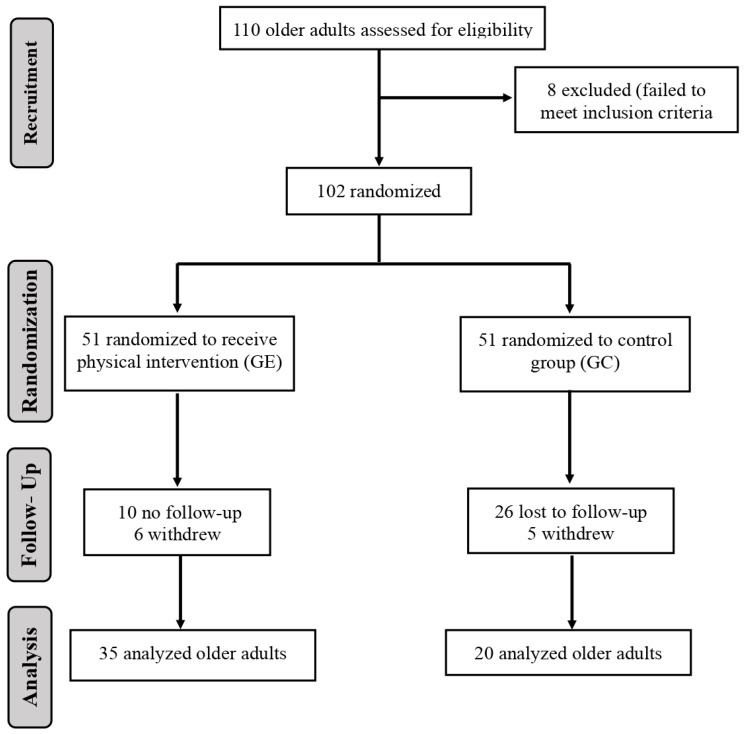
Subjects Flow diagram from initial contact through study completion.

**Figure 2 ijerph-20-02755-f002:**
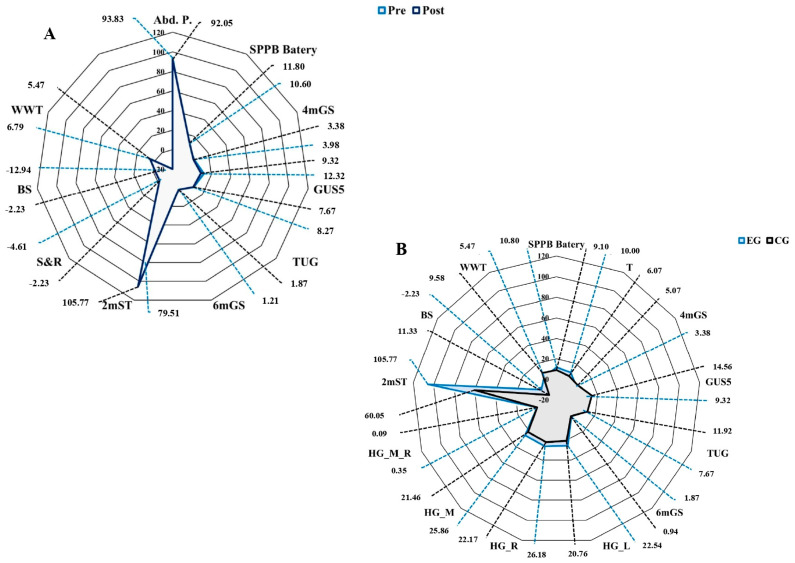
Spider chart showing the main differences in the functional capacity and physical capacity tests. Note: (**A**) EG. Comparison of the median significant values obtained for each test before and after the application of the MPTP. (**B**) Comparison between the experimental and control groups after the application of the MPTP program.

**Figure 3 ijerph-20-02755-f003:**
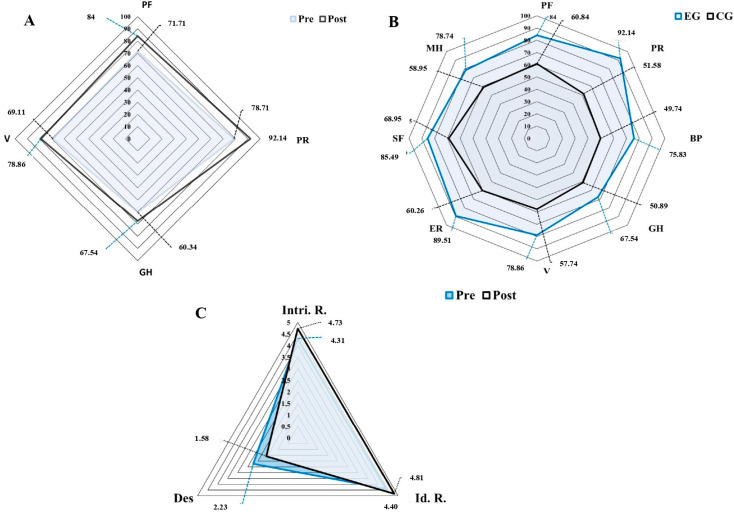
Spider chart accounts for quality of life and motivation to exercise before and after the MPTP in each of the groups. Note: (**A**) EG. Comparison of the mean significant values obtained for the SF-36 health questionnaire before and after the application of the MPTP; (**B**) Comparison between the experimental and control groups after applying the MPTP program in the SF-36 health questionnaire; (**C**) EG. Comparison of the mean significant values obtained for the BREQ-3 Questionnaire before and after the application of the MPTP.

**Table 1 ijerph-20-02755-t001:** Descriptive data of the sample according to the assigned training group.

	Total	EG	CG	*p*	d
	M ± SD (*n* = 55)	M ± SD (*n* = 35)	M ± SD (*n* = 20)		
Age (years)	69.42 ± 6.09	69.91 ± 5.83	68.55 ± 6.36	0.423	0.23
Height (cm)	156.17 ± 8.29	156.05 ± 8.56	156.37 ± 7.99	0.895	−0.04
Peso (kg)	74.77 ± 14.01	77.55 ± 14.19	75.15 ± 14.03	0.881	−0.04
IMC (Kg/m^2^)	30.54 ± 4.53	30.49 ± 4.55	30.63 ± 4.61	0.917	−0.03
Abdominal perimeter (cm)	94.93 ± 11.81	93.83 ± 12.08	96.60 ± 11.42	0.407	−0.23
% Fat	41.01 ± 11.52	40.88 ± 10.80	41.24 ± 12.97	0.913	−0.03
% Muscle mass	24.39 ± 5.54	24.85 ± 4.64	23.54 ± 4.33	0.318	0.29

Note: *n* = number of subjects; M = mean; SD = standard deviation; EG = experimental group; CG = control group; cm = centimeters; Kg = kilograms; kg/m^2^ = kilograms/square meters; d = Cohen’s d; *p* = significance level.

**Table 2 ijerph-20-02755-t002:** Training program by phases or blocks.

Phase 1 Strength Training (Week 1–9)					
B1: Neuromuscular/Adaptation		B2: Muscle Power		B3: Muscular Endurance	
Exercises	Load	Exercises	Load	Exercises	Load
	S: 2	Chest press	S: 3	Chest press	S: 3
Chest press	R: 6(10)	Leg extension	R: 5(10)	Leg extension	R: 10(12)
Leg extension	I: 75%MR	Rowing	I: 75%MR	Rowing	I: 70%MR
Rowing	C: 1-0-1	Leg press	C: x-1-1	Leg press	C: 1-0-1
Leg Press	RPE: 4–5	Triceps pull	RPE: 3–4	Triceps pull	RPE: 8–9
Hip abduction elastic band	RT: 1′	Hip abduction elastic band	RT: 2′	Hip abduction elastic band	RT: 2′
	ST: 25′		ST: 25′	Rises from a chair	TE: 25′
**Phase 2 Endurance Training (Week 10–18)**					
B1: Static gait		B2: Dynamic gait		B3: Dynamic gait/multidirection	
	S: 2		S: 2		S: 2
Chest press	R: 4(10)	Chest press	R: 4(10)	Chest press	R: 4(10)
Leg extension	I: 75%MR	Leg extension	I: 75%MR	Leg extension	I: 75%MR
Rowing	C: x-1-1	Rowing	C: x-1-1	Rowing	C: x-1-1
Leg Press	RPE: 3–4	Leg press	RPE: 3–4	Leg press	RPE: 3–4
	RT: 1′		RT: 1′		RT: 1′
	ST: 10′		ST: 10′		ST: 10′
Intermittent static gait	S: 3 × 6′	Intermittent dynamic gait	S: 3 × 6′	Dynamic gait with changes of direction	S: 3 × 5′
	WT: 10″		WT: 15″		WT: 20″
	RT: 20″		RT: 15″		RT: 10″
	Dn: 1:2		Dn: 1:1		Dn: 2:1
	P: 1′		P: 1′		P: 2:30′
	RPE: 6		RPE: 7		RPE: 7
	ST: 20′		ST: 20′		ST: 20′
**Phase 3 Flexibility and Balance (Week 19–27)**					
B1: Static balance-Flexibility		B2: Dynamic Balance/Flexibility		B3: Dual Task Dynamic Balance/Flexibility	
	S: 1		S: 1		S: 1
Chest press	R: 7(10)	Chest press	R: 7(10)	Chest press	R: 7(10)
Leg extension	I: 75%MR	Leg extension	I: 75%MR	Leg extension	I: 75%MR
Rowing	C: x-1-1	Rowing	C: x-1-1	Rowing	C: x-1-1
Leg Press	RPE: 3–4	Leg press	RPE: 3–4	Leg press	RPE: 3–4
	RT: 2′		RT: 2′		RT: 2′
	ST: 10′		ST: 10′		ST: 10′
Intermittent dynamic gait	S: 2 × 4′	Dynamic gait with changes of direction	S: 2 × 4′	Dynamic gait with changes of direction on the demarcated line and with balloon manipulation.	S: 2 × 4′
	WT: 10″		WT: 15″		WT: 15″
	RT: 20″		RT: 15″		RT: 15″
	Dn: 1:2		Dn: 1:1		Dn: 1:1
	P: 2 min		P: 2 min		P: 2 min
	RPE: 6		RPE: 7		RPE: 8
	ST: 10′		ST: 10′		ST: 10′
Hip abduction and static unipodal thrust Cross Balance Active stretching: Upper, lower limb and trunk.	S: 4	Bosu squat. Static gait in minitramp Stand up and sit down from the chair in semitandem Active stretching: upper limb, lower limb and trunk.	S: 4	Balance in bosu manipulating a balloon Straight march on demarcated line naming the vowels Active stretching: Upper, lower limb and trunk.	S: 4
	R: 15″		R: 15″		R: 15″
	RPE: 6–8		RPE: 6–8		RPE: 6–8
	RT: 1′		RT: 1′		RT: 1′
	ST: 20′		ST: 20′		ST: 20′

Note. B: block; S: series; R: repetitions; I: intensity; MR: maximum repetition; C: cadence; RPE; range of perceived exertion; RT: rest time; ST: spent time; WT: working time; Dn: density; P: pause.

**Table 3 ijerph-20-02755-t003:** Mean values obtained for functional capacity and physical capacity before and after MPTP in each of the groups.

	EG (*n* = 35)					CG (*n* = 20)					Post Intervention (*n* = 55)		
	Pre	Post	Δ (Pre_Post)	*p*	ES	Pre	Post	Δ (Pre_Post)	*p*	ES	Δ (CG_EG)	*p*	ES
	M ± SD	M ± SD	M ± SD			M ± SD	M ± SD	M ± SD			M ± SD		
Abd. P. (cm)	93.83 ± 12.08	92.05 ± 11.52	1.77 ± 0.56	0.034 *	0.08	96.60 ± 11.42	96.20 ± 12.22	0.40 ± −0.79	0.712	0.00	4.15 ± 0.70	0.215	0.03
Functional capacity													
SPPB	10.60 ± 1.67	11.80 ± 0.47	−1.20 ± 1.19	0.000 *	0.26	9.45 ± 2.63	9.10 ± 2.90	0.35 ± −0.27	0.578	0.01	−2.70 ± 2.43	0.000 **	0.36
Balance													
FT (sg)	10.00 ± 0.00	10.00 ± 0.00	0.00 ± 0.00	1.000	0.00	9.70 ± 1.34	9.75 ± 1.12	−0.05 ± 0.22	0.101	0.05	−0.25 ± 1.12	0.188	0.03
T (sg)	9.17 ± 0.17	10.00 ± 0.00	−0.83 ± 0.17	0.191	0.03	6.85 ± 4.51	6.07 ± 4.95	0.78 ± −0.44	0.299	0.02	−3.93 ± 4.95	0.000 **	0.30
ST (sg)	9.97 ± 2.13	10.00 ± 0.00	−0.03 ± 2.13	0.927	0.00	8.50 ± 3.66	9.50 ± 2.24	−1.00 ± 1.43	0.019 *	0.10	−0.50 ± 2.24	0.188	0.03
4 mGS (sg)	3.98 ± 1.34	3.38 ± 0.88	0.60 ± 0.46	0.001 **	0.20	4.17 ± 1.80	5.07 ± 1.85	−0.90 ± −0.06	0.000 **	0.25	1.69 ± 0.98	0.000 **	0.29
GUS5 (sg)	12.32 ± 4.53	9.32 ± 1.83	3.01 ± 2.70	0.000 **	0.32	14.67 ± 5.48	14.56 ± 4.76	0.11 ± 0.72	0.893	0.00	5.24 ± 2.94	0.000 **	0.39
TUG (sg)	8.27 ± 2.08	7.67 ± 1.85	0.61 ± 0.22	0.038 *	0.08	10.53 ± 6.56	11.92 ± 6.23	−1.40 ± 0.32	0.001 **	0.20	4.26 ± 4.38	0.000 **	0.20
6 mGS (mts/sg)	1.21 ± 0.35	1.87 ± 0.39	−0.66 ± −0.04	0.000 **	0.81	1.12 ± 0.30	0.94 ± 0.23	0.18 ± 0.07	0.003 **	0.64	−0.93 ± −0.15	0.000 **	0.64
Physical capacity													
HG_L (Kg)	25.40 ± 7.42	25.54 ± 6.91	−0.13 ± 0.51	0.831	0.00	19.68 ± 6.49	20.76 ± 7.37	−1.09 ± −0.88	0.197	0.03	−4.78 ± 0.46	0.020 *	0.10
HG_R (Kg)	26.06 ± 7.34	26.18 ± 6.55	−0.12 ± 0.79	0.847	0.00	21.22 ± 7.66	22.17 ± 7.59	−0.96 ± 0.07	0.238	0.03	−4.01 ± 1.04	0.044 *	0.07
HG_M (Kg)	27.73 ± 7.06	25.86 ± 6.55	1.87 ± 0.51	0.821	0.00	20.45 ± 6.89	21.46 ± 7.21	−1.02 ± −0.32	0.169	0.04	−4.40 ± 0.66	0.025 *	0.09
HG_M_R (Kg/mass kg)	0.25 ± 0.09	0.35 ± 0.08	−0.10 ± 0.01	0.934	0.00	0.28 ± 0.09	0.09 ± 0.36	0.19 ± −0.27	0.222	0.03	−0.26 ± 0.27	0.025 *	0.14
2 mST (steps)	79.51 ± 22.42	105.77 ± 20.11	−26.26 ± 2.32	0.000 **	0.48	62.25 ± 28.52	60.05 ± 28.46	2.20 ± 0.06	0.658	0.00	−45.72 ± 8.35	0.000 **	0.48
FEV_1_ (L/s)	1.86 ± 0.78	1.86 ± 0.61	0.00 ± 0.17	0.943	0.00	1.54 ± 0.48	1.53 ± 0.53	0.01 ± −0.04	0.951	0.00	−0.33 ± −0.08	0.049 *	0.05
Flexibility													
SR (cm)	−4.61 ± 7.72	−2.23 ± 5.95	−2.39 ± 1.77	0.047 *	0.07	−4.89 ± 9.50	−5.26 ± 9.63	0.37 ± −0.13	0.818	0.00	−3.03 ± 3.68	0.158	0.04
BS (cm)	−12.94 ± 12.06	−2.23 ± 5.95	−10.71 ± 6.11	0.000 **	0.36	−5.73 ± 8.46	−11.33 ± 11.93	5.60 ± −3.47	0.316	0.02	−9.10 ± 5.98	0.003 **	0.18
WWT (sg)	6.79 ± 2.29	5.47 ± 1.43	1.32 ± 0.86	0.000 **	0.77	11.17 ± 5.74	9.58 ± 4.02	1.59 ± 1.72	0.44	0.07	4.11 ± 2.59	0.000 **	0.36

Note: M = mean; SD = standard deviation; sec = seconds; cm = centimeters; meters = meters; Kg = kilograms; * = statistical difference *p* < 0.05; ** = statistical difference *p* < 0.01; ES = effect size; Δ = difference; Abd. P. = abdominal perimeter; SPBB = short physical performance battery; FT = balance test feet together; T = tandem balance; ST = semi-tandem balance; 4 mGS = test gait speed of 4 m; GUS5 = test of getting up and sitting on a chair 5 times; TUG = timed up and go; 6 mGS = 6 m gait speed test; HG_L = left hand grip strength; HG_R = right hand grip strength; HG_M = mean manual grip strength; HG_M_R = relative hand grip strength; 2 Mst = 2 min step walk in place two minutes; FEV_1_ = maximum expired volume in first second.

**Table 4 ijerph-20-02755-t004:** Mean values obtained for quality of life and motivation for exercise before and after the MPTP in each of the groups.

	EG (*n* = 35)					CG (*n* = 20)					Post Intervention (*n* = 55)		
	Pre	Post	Δ (Pre_Post)	*p*	ES	Pre	Post	Δ (Pre_Post)	*p*	ES	Δ (CG_EG)	*p*	ES
	M ± SD	M ± SD	M ± SD			M ± SD	M ± SD	M ± SD			M ± SD		
Quality of Life													
SF-36													
PF	71.71 ± 23.79	84.00 ± 16.97	−12.29 ± 6.82	0.000 **	0.26	56.75 ± 27.69	60.84 ± 27.53	−4.09 ± 0.16	0.103	0.05	−23.2 ± 10.6	0.000 **	0.26
PR	78.71 ± 33.83	92.14 ± 23.30	−13.43 ± 10.53	0.039 *	0.08	55.00 ± 43.38	51.58 ± 36.45	3.42 ± 6.92	0.466	0.01	−40.6 ± 13.2	0.000 **	0.32
BP	69.11 ± 23.54	75.83 ± 22.56	−6.71 ± 0.98	0.127	0.04	58.95 ± 29.25	49.74 ± 25.30	9.21 ± 3.95	0.090	0.05	−26.1 ± 2.74	0.000 **	0.23
GH	60.34 ± 17.89	67.54 ± 18.88	−7.20 ± −0.99	0.046 *	0.67	52.80 ± 19.66	50.89 ± 17.64	1.91 ± 2.02	0.669	0.05	−16.6 ± −1.23	0.003 **	0.16
V	69.11 ± 19.26	78.86 ± 17.74	−9.74 ± 1.52	0.012 *	0.12	58.75 ± 24.11	57.74 ± 18.21	1.01 ± 5.89	0.694	0.00	−21.1 ± 0.47	0.000 **	0.25
ER	80.97 ± 32.65	89.51 ± 27.77	−8.54 ± 4.88	0.229	0.03	45.00 ± 46.26	60.26 ± 34.74	−15.26 ± 11.51	0.182	0.03	−29.3 ± 6.98	0.001 **	0.18
SF	80.86 ± 20.34	85.49 ± 17.72	−4.63 ± 2.62	0.218	0.03	65.80 ± 24.19	68.95 ± 26.87	−3.15 ± −2.68	0.648	0.00	−16.5 ± 9.15	0.009 **	0.13
MH	77.83 ± 19.09	78.74 ± 14.93	−0.91 ± 4.17	0.797	0.00	57.20 ± 24.93	58.95 ± 20.26	−1.75 ± 4.67	0.760	0.00	−19.8 ± 5.33	0.000 **	0.24
Motivation for exercise													
Intri. R.	4.31 ± 0.95	4.73 ± 0.37	−0.42 ± 0.57	0.006 **	0.14	3.80 ± 1.33	3.70 ± 1.25	0.11 ± 0.08	0.591	0.01	−1.03 ± 0.88	0.000 **	0.28
Inte. R.	4.39 ± 0.90	4.17 ± 0.77	0.22 ± 0.12	0.197	0.03	3.83 ± 1.51	3.66 ± 1.26	0.17 ± 0.26	0.471	0.01	−0.51 ± 0.49	0.068	0.06
Id. R.	4.40 ± 1.01	4.81 ± 0.38	−0.41 ± 0.64	0.009 **	0.12	4.37 ± 1.12	4.04 ± 1.07	0.33 ± 0.05	0.915	0.01	−0.77 ± 0.69	0.000 **	0.22
Intro. R.	3.06 ± 1.14	2.85 ± 1.18	0.21 ± −0.04	0.328	0.02	2.79 ± 1.16	2.60 ± 1.16	0.19 ± 0.01	0.488	0.01	−0.26 ± −0.03	0.439	0.01
Ext. R.	2.18 ± 1.20	1.81 ± 1.10	0.37 ± 0.11	0.082	0.06	2.15 ± 1.01	1.86 ± 0.77	0.29 ± 0.24	0.295	0.02	0.049 ± −0.33	0.862	0.00
Des	2.23 ± 1.45	1.58 ± 0.77	0.65 ± 0.68	0.003 **	0.15	1.51 ± 0.59	1.91 ± 1.00	−0.41 ± −0.41	0.156	0.04	0.33 ± 0.23	0.171	0.04
RAI	13.79 ± 6.64	14.17 ± 7.03	−0.37 ± −0.39	0.714	0.00	10.62 ± 6.64	11.61 ± 6.09	−0.99 ± 0.55	0.462	0.01	−2.56 ± −0.94	0.179	0.03
DAI	11.64 ± 3.85	11.64 ± 4.00	−0.01 ± −0.15	0.992	0.00	9.50 ± 4.12	9.87 ± 3.87	−0.37 ± 0.24	0.621	0.00	−1.77 ± −0.13	0.116	0.05

Note: M = mean; SD = standard deviation; sec = seconds; cm = centimeters; meters = meters; Kg = kilograms; * = statistical difference *p* < 0.05; ** = statistical difference *p* < 0.01; ES= effect size; Δ = difference; Abd.P. = abdominal perimeter; SPBB = short physical performance battery; FT = balance test feet together; T = tandem balance; ST = semi-tandem balance; 4 mGSs = test gait speed of 4 m; GUS5 = test of getting up and sitting on a chair 5 times; TUG = timed up and go; 6 mGS = 6 m gait speed test; HG_L = left hand grip strength; HG_R = right hand grip strength; HG_M = mean manual grip strength; HG_M_R = relative hand grip strength; 2 Mst = 2 min step walk in place two minutes; FEV_1_ = maximum expired volume in first second. More specifically, after the older adults underwent the MPTP, it was possible to observe how the mean score for each and every one of the dimensions of the SF-36 questionnaire was higher in the EG than in the CG, experiencing an increase in the score levels of 38.06% for the PF, 78.64% for PR, 52.46% in the BP dimension, 32.71% in the GH dimension, 36.58% in V, 48.54% for the ER dimension, 23.99% in SF and 33.58 % for MH dimension. In addition, in this same analysis, when studying and comparing the scores obtained by the EG before and after the application of the EM, differences were observed in the dimensions of PF (*p* = 0.000), PR (0.039), GH (*p* = 0.046), and vitality (*p* = 0.012). In all cases, this EG increased the mean scores obtained (Table 3). Finally, in the case of the CG, when comparing their scores before and after this period of EM, no significant differences were observed in any of the scores obtained (*p* > 0.05). To conclude, in this same Table 4, when comparing the results obtained in the BREQ-3 questionnaire on the motivation towards physical exercise, it was possible to verify how when comparing the values of the mean score for the dimensions of Intri.R. (*p* = 0.000) and Id. R. (*p* = 0.000) there were significant differences between the EG and CG after the ME period. More specifically, the EG showed higher values in each and every one of these dimensions with a percentage of 27.97% and 9.29%, respectively. However, along the same lines as the results presented above, when comparing the values of the mean scores obtained by the EG before and after the EM period, significant differences were observed for the dimensions of Intri.R. (*p* = 0.006), Id.R. (*p* = 0.009) and demotivation (*p* = 0.003). These differences showed an increase in the mean values of 9.78% for Intr.R. and 9.28% for Id.R., while for the DE dimension a decrease of 29.15% in the initial value obtained was observed.

## Data Availability

The datasets used during the current study are available from the corresponding authors on reasonable request.

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
