# Peer review of "Multicomponent Training in Progressive Phases Improves Functional Capacity, Physical Capacity, Quality of Life, and Exercise Motivation in Community-Dwelling Older Adults: A Randomized Clinical Trial"

_ijerph, 2023, doi:10.3390/ijerph20032755_

Round 1
Reviewer 1 Report
The manuscript presents the results of evaluation on effects of MPTP on older adults’ functionality, quality of life (QoL) and motivation to exercise (EM). The manuscript was well constructed and easy to follow. The research could also provide important information for the development of health promotion strategies in elderly groups. I have some comments as follow:
Since this research has three dependent variables including functionality, quality of life (QoL) and motivation to exercise (EM), the introduction part should clearly discuss how MPTP could affect each single dependent variable. I am particularly interested how MPTP could maintain older adults’ motivation to do exercise and enhance quality of life.
In the introduction, please clearly explain the definitions and meanings of older adults’ quality of life, and which aspects of QoL among older adults could be assumably enhanced by MPTP.
Please add an explanation on why the assessment of functional capacity, physical capacity, quality of life, and body composition were carried out at weeks 1, 9, 18, and 27, any why the assessment of motivation to exercise was assessed weeks 1 and 27.
Please explain why many participants in a control group did not continue participating in the research.
Why don’t report participants’ gender in both group? This is because gender could also affect those dependent variables as well.
Author Response
Comments and Suggestions for Authors
The manuscript presents the results of evaluation on effects of MPTP on older adults’ functionality, quality of life (QoL) and motivation to exercise (EM). The manuscript was well constructed and easy to follow. The research could also provide important information for the development of health promotion strategies in elderly groups. I have some comments as follow:
First of all, we would like to express our gratitude to the reviewer for their contributions. These recommendations have allowed us to improve the writing style and make the manuscript more straightforward. We hope the changes are appropriate.
All the answers and changes made in the manuscript will be detailed according to the reviewer’s questions or suggestions.
Since this research has three dependent variables including functionality, quality of life (QoL) and motivation to exercise (EM), the introduction part should clearly discuss how MPTP could affect each single dependent variable. I am particularly interested how MPTP could maintain older adults’ motivation to do exercise and enhance quality of life.
Thank you very much for this contribution. We have proceeded to incorporate what was requested for the introduction.
In the introduction, please clearly explain the definitions and meanings of older adults’ quality of life, and which aspects of QoL among older adults could be assumably enhanced by MPTP.
Again, thanks very much to the reviewer for this contribution. We have proceeded to incorporate it in the introduction section.
Please add an explanation on why the assessment of functional capacity, physical capacity, quality of life, and body composition were carried out at weeks 1, 9, 18, and 27, any why the assessment of motivation to exercise was assessed weeks 1 and 27.
Thank you very much for this contribution. We have proceeded to incorporate what was requested for the methodology section explaining why the tests were applied in those weeks.
Please explain why many participants in a control group did not continue participating in the research.
Thank you very much for this contribution. We have proceeded to explain why a part of the participants of each group was lost.
Why don’t report participants’ gender in both group? This is because gender could also affect those dependent variables as well.
The subjects were not all of the same sex; however, the number of older males was very low (n=11; NGC=4 and NGE=9). Therefore, after careful analysis, the authors decided not to take into account the difference by gender. It is not methodologically correct to carry out an inferential statistical analysis with this small number of subjects and, of course, to speak of significant differences. Furthermore, if we had opted for a gender analysis, a problem of validity of the statistical conclusion would have been created. For all these reasons, we decided not to take into account this differentiation by gender.
Of course, we would have liked to have a greater number of male participants, but during the selection and recruitment process, the men showed no interest in participating in the project to be developed, much less after being aware of the number of weeks and days that they had to commit to carry out. We know that this is a possible limitation of the study, but the selection of the sample in any investigation is a delicate and complex process in which, as researchers, on many occasions we cannot count on the number of participants that we would like to do. We hope that the reviewers are aware of this and value, on the other hand, the potential that the application of this type of intervention presents in population groups of these ages, as well as the rigor with which the entire process has been developed from the beginning. from our study research.
Lastly, the authors would like to thank the reviewer for the suggestions. We believe that the article has improved with all the modifications made. I hope that we have addressed all the reviewer's suggestions or queries.

Reviewer 2 Report
A well done experiment, needs only minor editing to improve readability.
Row 80: It is unclear why a period was inserted,may be some typo? or something is missing?
row 93: give median and range for age, subjects were all of the same sex? it seems important to state if they were male or female given the Choen's D in last row of table 1 (speaking of it: was a correction factor used given the closeness to 50 of the records?)
figure 1 shows a great difference between dropouts, why close to 50% of CG subjects were lost to follow up? It's important to know if they were lost for "logistic and administrative" reasons or for diseases and death.
Table3 and 4 are very useful, but also not easy to read; i suggest to keep it but to consider the possibility to use also a figure (spider graph may be) to show only the major improvements reported in table
For the discussion a comment on the possibility that working together (10-15 persons groups) may contribute to quality of life and program compliance will be useful.
Author Response
Comments and Suggestions for Authors
A well done experiment, needs only minor editing to improve readability.
First of all, we would like to express our gratitude to the reviewer for their contributions. These recommendations have allowed us to improve the writing style and make the manuscript more straightforward. We hope the changes are appropriate.
All the answers and changes made in the manuscript will be detailed according to the reviewer’s questions or suggestions.
Row 80: It is unclear why a period was inserted, may be some typo? or something is missing?
First of all, thank very much for your suggestions. We have proceeded to modify the mistake.
row 93: give median and range for age, subjects were all of the same sex? it seems important to state if they were male or female given the Choen's D in last row of table 1 (speaking of it: was a correction factor used given the closeness to 50 of the records?)
Again, thank very much for your suggestion. The authors will detail all modifications point by point:
- We have proceeded to complete the information requested by the reviewer on line 99.
- The subjects were not all of the same sex; however, the number of older males was very low (n=11; NGC=4 and NGE=9). That is why, after analyzing it carefully, the authors decided not to take into account the difference by gender. It is not methodologically correct to perform an inferential statistical analysis with this small number of subjects and, of course, talk about significant differences. Moreover, if we had opted for an analysis by gender, this would have created a problem of validity of the statistical conclusion. For all these reasons, we decided not to take into account this differentiation by gender.
Of course, we would have liked to have had a higher number of male participants, but during the selection and recruitment process, the men showed no interest in participating in the project to be developed, much less after being aware of the number of weeks and days that they had to commit to carry out. We know that this is a possible limitation of the study, but the selection of the sample in any investigation is a delicate and complex process in which as researchers on many occasions we cannot count on the number of participants that we would like to do. We hope that the reviewers are aware of this and value, on the other hand, the potential that the application of this type of intervention presents in population groups of these ages, as well as the rigor with which the entire process has been developed since the beginning of our study research.
- On the other hand, regarding the doubt expressed by the reviewer about Cohen's D, the authors are aware of the problems and overestimation that it can occur in the calculation of the effect size for small samples (n<50). In this case, we followed the recommendations given by the he specific literature on the subject. As an example, we can cite the following documents:
Coe, R., y Merino-Soto, C. (2003). Magnitud del efecto: Una guía para investigadores y usuarios. Revista de Psicología de la PUCP, XX1(1), 145-177.
Cohen J. (1998). Statistical power analysis for the behavioral sciences (2nd edition). Routledge
Ellis, P. (2010). The essential guide to effect sizes: Statistical power, Meta-Analysis and the interpretation of research results (1st edition). Cambridge University Press.
figure 1 shows a great difference between dropouts, why close to 50% of CG subjects were lost to follow up? It's important to know if they were lost for "logistic and administrative" reasons or for diseases and death.
Again, thanks very much to the reviewer for this contribution. We have proceeded to explain why a part of the participants of each group was lost in the sample size section, lines 138 to 142.
Table3 and 4 are very useful, but also not easy to read; i suggest to keep it but to consider the possibility to use also a figure (spider graph may be) to show only the major improvements reported in table
Thanks very much to the reviewer for the suggestion. In accordance with your suggestion, we have proceeded to include the figure (spider graph).
For the discussion a comment on the possibility that working together (10-15 persons groups) may contribute to quality of life and program compliance will be useful.
Again, thank you very much for this contribution. We have proceeded to incorporate what was requested for the discussion section in lines 696 to 700.
Lastly, the authors would like to thank the reviewer for the suggestions. We believe that the article has improved with all the modifications made. I hope that we have addressed all the reviewer's suggestions or queries.

Reviewer 3 Report
Line 81 - There is a dot (.) out of place.
Line 156 - There are a dot (.) and a parenthesis [)]out of place.
Table 4 - The phrase "Motivación por el ejercicio" is in Spanish.
Author Response
Comments and Suggestions for Authors
First of all, we would like to express our gratitude to the reviewer for their contributions. These recommendations have allowed us to improve the writing style and make the manuscript more straightforward. We hope the changes are appropriate.
All the answers and changes made in the manuscript will be detailed according to the reviewer’s questions or issues.
Line 81 - There is a dot (.) out of place.
First of all, thank very much for your suggestions. We have proceeded to modify the mistake.
Line 156 - There are a dot (.) and a parenthesis [)]out of place.
Again, thanks very much for your appreciation. The authors proceeded to correct the mistakes.
Table 4 - The phrase "Motivación por el ejercicio" is in Spanish.
Lastly, sorry for the mistake. We proceeded to correct it. Thanks very much to the reviewer for the corrections.
Lastly, the authors would like to thank the reviewer for the suggestions. We believe that the article has improved with all the modifications made. I hope that we have addressed all the reviewer's suggestions or queries.
